# Research on Profit-Sharing Mechanism of IPD Projects Considering Multidimensional Fairness Preferences and BIM

**Lunyan Wang** [1,2]**, Mengyu Tao** [3]**, Xiaowei An** [3,*] **and Guanghua Dong** [3]

1    College of Water Resources, North China University of Water Resource and Electric Power, Zhengzhou 450046, China; wanglunyan@ncwu.edu.cn
2    Henan Key Laboratory of Water Environment Simulation and Treatment, Zhengzhou 450046, China
3    School of Water Conservancy, North China University of Water Resource and Electric Power, Zhengzhou 450046, China; taomengyu123@foxmail.com (M.T.); dongguanghua@ncwu.edu.cn (G.D.)
*    Correspondence: anxiaowei@ncwu.edu.cn

**Abstract:** The integration of building information modeling (BIM) and the integrated project delivery (IPD) mode effectively promotes collaboration among project members and enhances project profitability. However, the issue of profit sharing significantly impacts the successful implementation of IPD projects. To enhance the profit-sharing mechanism of IPD projects and ensure their smooth implementation, a game analysis model of profit sharing in IPD projects was established based on the Stackelberg game theory, taking into account the multidimensional fair preferences of the participants and the application of BIM technology. Through simulation, the impact of various parameters of participants on output utility, total revenue, and sharing coefficient in IPD projects was analyzed. The results show that: (1) participants achieve their highest output utility and total revenue under vertical–horizontal fairness preferences; (2) under vertical fairness preferences, the profit sharing coefficient is the highest, while the output utility and total revenue are the lowest; (3) although the output utility and total revenue of participants under horizontal fairness preferences exceed those under neutral fairness preferences, the profit-sharing coefficient is lower; (4) the output utility, the total revenue, and the profit-sharing coefficient of the participants all increase with the increase in effort utility value and decrease with the increase in the effort cost coefficient and the risk avoidance coefficient. The research findings provide valuable theoretical support for the profit sharing of IPD projects, thereby further promoting the advancement and implementation of the IPD model.

**Keywords:** integrated project delivery model; building information modeling; profit sharing; multidimensional fairness preferences; simulation



## 1. Introduction

Traditional project delivery methods suffer from performance issues due to their phased structure [1]. Integrated project delivery (IPD), as a new delivery model, has been widely used, having the advantage of strong team integration compared with the traditional delivery model [2]. In the IPD model, there are seven stages in the construction project, logical identified as conceptualization, standards design, comprehensive design, execution documents, agency final buyout, construction, and finally, the closeout stage [1].The IPD model integrates various construction elements into a cooperation process, permeating throughout the entire lifecycle of the project [3] and reducing misunderstandings, redundant works, and disputes caused by insufficient collaboration awareness among the participants [4]. Furthermore, the IPD model emphasizes the early involvement of key participants, multi-party contracts, shared risk and reward, as well as the dissemination of information and transparency. This introduction of the model forges a more efficient, collaborative, and high-performance project delivery environment, thereby furnishing a platform to facilitate effective communication amongst project teams and fostering a scenario conducive to mutual gains for all parties involved [5]. The Building Information

Model (BIM) is frequently associated with a tool, software, methodology, representative model, project simulation, revolutionary technology, or a modern concept used to generate an image and marketing [6]. As a new tool for design, construction, and management, BIM technology allows for a better understanding of the project information (for designing, building, and operating) through powerful visualization, information integration, and automation, effectively addressing the challenges of information management [7]. At the same time, BIM technology enables the real-time exchange of engineering information among project stakeholders and facilitates collaborative efforts through the creation of virtual building models and the resolution of compatibility issues. This capability promotes the exchange and interoperability of information throughout the engineering construction process [8] and bolsters the cultivation of trust and cooperation required by IPD [9], thus exerting a positive impact on the implementation of IPD [10]. The application of BIM provides technical support for the implementation of IPD projects. At the same time, the IPD model improves the collaborative environment required for BIM implementation, providing organizational management means for the implementation of BIM [11]. Therefore, the coupling of BIM and IPD effectively promotes integration and interaction among project members [12], thereby enhancing team efficiency and resource conservation and optimizing project outcomes [13]. Moreover, in the context of BIM and IPD collaboration, the core idea lies in the sharing of profits and the distribution of risks [14], as a successful IPD project relies on a reasonable profit-sharing mechanism [15]. However, in an IPD project, each participant is an independent economic entity and aims to maximize their own profits. In the interest alliance consisting of IPD project members, if each participant fails to obtain satisfactory sharing of profits, their participation enthusiasm will be affected, which will eventually lead to the disintegration of the whole interest alliance [16]. Therefore, a scientific and fair profit-sharing mechanism is crucial for the normal operation of the alliance, as well as for achieving resource complementarity and the sharing of profits among alliance members [17]. Thus, a fair and reasonable profit-sharing mechanism is the basis for long-term and stable cooperation among all participants in IPD projects. It is also the guarantee for the efficient completion of the IPD projects and the key to the successful coupling of BIM and the IPD mode.

In recent years, researchers studying profit sharing have mainly focused on the analysis of influencing factors and the selection of sharing methods. In terms of influencing factor analysis, Du et al. [18] pointed out five key factors that influence profit sharing from a private sector perspective: risk sharing, financing capacity, investment, management ability, and effort level. Dai et al. [19] constructed a two-stage profit sharing model with two types of communication structures, exploring the impact of communication structure constraints and task completion quality on profit sharing in logistics alliances. Zhang and Li [20] proposed a risk/reward compensation model to incentivize and adjust the goals of all participants to optimize the profit sharing of IPD projects. The selection of profit-sharing methods mainly includes: the Generic Function Model Method, the simplified Minimum Cost-Remaining Savings (MCRS) method, the Nash negotiation model, the Core method, and the Shapley value method [21]. Among them, there is a focus on the selection of evolutionary games and the improved Shapley value [22]. Utilizing the Cobb–Douglas function, Wang [23] constructed a profit-sharing model between general contractors and subcontractors in construction projects, analyzing the issue of profit sharing for both one-time and multiple collaborations between the parties under the decision-making frameworks of self-interest and collectivism. Huang et al. [24] established a Stackelberg game model of dynamic alliance under government regulation, analyzing the optimal alliance strategy of enterprises, and combined the Shapley value method to coordinate the sharing of optimal alliance profits. Han and Yang [25] developed a tripartite evolutionary game model of "government-member firm A-member firm B" and analyzed the profit sharing and stability among alliance members. Based on evolutionary game theory, Hosseini et al. [26] established a profit-sharing model for supply chains and analyzed the effect of members' dynamic strategy choices on their shared profits. An improved Shapley model for the

equitable sharing of cooperative profits was proposed by Song et al. [27], based on the efficiency measurement model of fuzzy DEA by replacing the marginal efficiency of each member with the value of marginal efficiency contribution. Based on the risk assessment model of the comprehensive fuzzy evaluation method (BWM-FCE), Wang et al. [28] combined capital input, contribution, and project participation, optimizing the Shapley value method to obtain a fairer and more reasonable profit-sharing system. In the aforementioned research approaches for profit sharing, the core concept of the General Function Modeling Method is to express the relationship between inputs and outputs through a mathematical function. This method facilitates a better understanding of a system's behavioral patterns and exhibits broad applicability. However, in complex system behaviors, it may not be possible to accurately describe them using a single mathematical function. The simplified MCRS method aims to enhance project benefits by optimizing resource sharing to minimize costs and maximize savings. It offers advantages such as feasibility and comprehensive cost-savings consideration. However, this approach inevitably involves subjective judgments in estimating the weights of costs and savings. The Nash bargaining model achieves balanced resource sharing by incorporating utility functions and strategies of project participants. This method provides enhanced insights into participants' behavioral motivations and potential negotiation outcomes. However, in its model construction, the emphasis on participants' pursuit of self-interest might overlook the aspect of equitable sharing. Core stability, as a form of cooperative game theory, underscores the inseparability of small groups from cooperation stability. While this method ensures the stability and satisfaction of participants, it does not always have a unique or existing solution in its calculations. The Shapley value method quantifies participants' contributions across various cooperative scenarios to determine the shared profits received by each party. While this approach possesses a degree of fairness, its computational complexity is significant due to the consideration of all possible cooperative scenarios. The Stackelberg game is employed to depict a scenario in which one participant makes decisions before another participant. It aids in simulating real-world leader–follower relationships and offers clear game equilibrium solutions. In the methodologies employed for profit-sharing mechanism research, the General Function Modeling Method describes an equitable game relationship without a predetermined sequence or role allocation. The simplified MCRS method, however, simplifies matters and places greater emphasis on cost savings. The Nash bargaining model, Core stability method, and Shapley value method focus on individual power distribution and game equilibrium within cooperative games. The Stackelberg game is applicable to leader–follower scenarios, highlighting first-mover advantage and strategic influence.

In the research on profit sharing in the IPD project, Li et al. [29] proposed a synergistic system framework for IPD projects using the data envelopment algorithm (DEA), which used the input–output efficiency of participants as the main criterion for benefit allocation. By considering the degree of participation of alliance members in IPD projects and the uncertainty of resource input, Guo et al. [30] established a Shapley value based on the Choquet integral to optimize profit sharing among IPD project stakeholders. Based on asymmetric information game theory and principal–agent theory, Guo et al. [31] explored the effect of effort level on the profit sharing of IPD projects and derived strategies of each participant in terms of effort level factors. Using a quadratic programming model based on a fuzzy cooperative game, Guo and Wang [32] considered different weights of IPD project participants and the efficiency of their participation in order to construct a fair and efficient profit-sharing scheme. A large body of research has provided a reference basis for the profit sharing of IPD projects, However, there is a lack of research concerning the impact of participants' fairness psychology on profit sharing.

Fairness psychology as a preference for behavioral choice has been included in various studies by many scholars. An et al. [33] believed that the process of optimizing profit sharing should take into account not only the rational behavior of the subject, but also the irrational behavior of the stakeholders' fairness concern. Considering the case of manufacturers with fairness preference and retailers with fairness neutrality, Qin et al. [34]

constructed a supply chain cost allocation model based on the Stackelberg model and determined the range of cost allocation ratios and optimal solutions. In the case of manufacturers with fairness preferences, Jiang et al. [35] analyzed the effect of manufacturers' fairness preferences on product prices, manufacturers' profits, suppliers' profits, and overall profits. Xu et al. [36] developed a principal–agent model with a fairness preference and showed that the fairness preference coefficient has a significant effect on profit sharing. Based on the F-S theory, Zou et al. [37] established a fairness preference model to elucidate the impact of fairness preferences on profit sharing, extending the profit-sharing theoretical model of the marketization of rural collectively owned commercial construction land. Lu et al. [38] developed an improved profit-sharing model for a two-level supply chain consisting of a Logistics Service Integrator (LSI) and several Functional Logistics Service Providers (FLSP). They discussed the impacts of inequity aversion and the number of members with inequity aversion to profit sharing. Wang [39] selected the Nash negotiation solution of entrepreneurial firms and venture capitalists as the reference point of fairness preference and constructed a principal–agent model of project investment in which firms have a fairness preference. The impact of fairness preferences on project profit sharing and the effort levels of both sides was studied. Based on the Fairness Preference theory, Jiang [40] established a profit-sharing model between green manufacturers and suppliers and analyzed the impact of fairness preferences under asymmetric information conditions on supplier green innovation, profit-sharing ratios, fixed subsidies, and the optimal utility of manufacturers. Zhao [41] introduced the theory of Fair Concern and formulated a Stackelberg game model between the design and construction parties. Taking into account the comprehensive effects of envy utility and sympathy utility for the design party, the study conducted an analysis on the optimal allocation ratio of earnings within the consortium and determined the optimal effort levels for both the design and construction parties. In IPD projects, fewer studies have considered the impact of fairness preference on profit sharing; however, the fairness preferences of project members directly affect their efforts [42] and the synergy between members [43]. Therefore, fairness preference is an important factor to consider in the research of profit-sharing mechanisms in IPD projects.

The above research indicates that various methods and models for dynamic alliance profit sharing have been developed. However, these studies primarily emphasize the profit-sharing mechanism within manufacturing supply chains, neglecting the impact of team collaboration and BIM technology on project profit sharing in the IPD mode. Furthermore, in the research conducted in the relevant field, a majority of profit sharing in the IPD mode lacks analysis from the perspective of project participants, failing to cater to the needs of all involved parties. Simultaneously, in the construction of profit-sharing methods and models, the majority of research has focused on Shapley values and their modifications utilizing certain profit-sharing elements. However, these methods are relatively simplistic and lack thoroughness. Lastly, fairness psychology, as a preference in behavioral choices, influences team collaboration among project members, yet there is a lack of research on its role in profit-sharing mechanisms in the IPD mode. Thus, based on Stackelberg game theory and considering the application of BIM technology and the horizontal fairness preference, the vertical fairness preference, as well as the vertical–horizontal fairness preference of the participants, a profit-sharing model of an IPD project is developed to investigate its profit-sharing mechanism. It can provide support for profit sharing in IPD projects, enhance cooperation between all parties, and help the promotion and successful implementation of the IPD model. The objectives and innovations of this paper are as follows:

(1) Based on Stackelberg game theory, with the project owner as the "leader" and the participants as the "followers" within the context of the IPD mode, the mutual impact of decisions between the owner and participants is analyzed.

(2) Considering the application of BIM technology by participants and the impact of fairness preferences on the overall project revenue, the horizontal fairness preferences, vertical fairness preferences, and vertical–horizontal fairness preferences of participants are introduced to construct a profit-sharing model for IPD projects.

(3)    Through simulation, explore the impact of participants' fairness preference intensity, effort utility value, effort cost coefficient, and risk avoidance coefficient on their output utility, shared coefficient, and total revenue. The objective is to encourage the project owner to establish rational cooperation arrangements and stimulate active engagement from all participants in project construction, ensuring cohesive collaboration among members and the smooth completion of IPD projects.

The remainder of this paper is structured as follows: Section 2 focuses on the construction of the profit-sharing model for IPD projects. It discusses the fundamental assumptions required for model construction and solves for the optimal model solution under the multidimensional fairness preferences of the participants. Section 3 involves simulation analysis, exploring how participants' fairness preference intensity, effort cost coefficient, effort utility value, and risk aversion coefficient impact output utility, benefit distribution coefficients, and overall returns under multidimensional fairness preferences. Section 4 is a conclusion. It discusses the results of the simulation analysis, presents relevant recommendations and strategies for project owners and participants in IPD projects, and explores the limitations of the study.

## 2. Development of IPD Projects' Profit-Sharing Model

The Stackelberg game is a type of strategic game model within game theory that highlights the impact of information asymmetry and sequential decision-making on game outcomes. In the Stackelberg game model, two distinct groups, namely the "leader" and the "followers", coexist, both driven by the goal of maximizing their individual interests. The leader formulates his or her decisions in advance, followed by the followers making their decisions based on the leader's choices. Consequently, the leader can exert influence over the followers by devising strategic choices, aiming to achieve the maximum benefit for the project. In order to enhance the profit-sharing mechanism in IPD projects and foster collaboration among all project stakeholders, a study on the IPD project profit-sharing mechanism was conducted. Firstly, the basic assumptions of the model were established based on the principles of Stackelberg game theory. The model was then constructed through an analysis of participants' output utility, effort cost, fair utility, and risk cost. Secondly, the study introduced participants' multidimensional fairness preferences and proceeded to determine the optimal sharing coefficients for each dimension of fairness preference within the model. Finally, employing simulation, an analysis was conducted to evaluate the impact of various parameters. The detailed procedural steps are illustrated in Figure 1.

### 2.1. Basic Assumptions

**Assumption 1:** *According to the core stakeholder theory [44], as well as an analysis of the contract clause C195 in AIA (American Institute of Architects), the Primary Team Members (PTMs) of the IPD mode typically include the owner, architect, general contractor, and other stakeholders [45]. Furthermore, early participants who significantly impact project design and cost are considered the most valuable. Therefore, considering the participants involved in profit sharing of IPD projects as the core stakeholders, the primary ones include the owner, architect, and contractor.*

**Assumption 2:** *In Integrated project delivery (IPD) mode, the owner forms a project alliance by selecting partners. Typically, the owner, architect, and general contractor will enter into multi-party agreements, while the other parties (e.g., consultants, suppliers, etc.) will enter into sub-contract agreements with the above three parties [31]. Therefore, in IPD projects with multi-party contract structures, the owner, as the project initiator, holds the ownership of the project, has higher dominant power, and is often the leader of the IPD projects, while the architect and contractor, as the project participants, are often the followers. Therefore, the profit sharing of IPD projects can be regarded as a Stackelberg game dominated by the owner.*

**Assumption 3:** *The IPD model adopts a comprehensive and cooperative approach, enhancing project returns through efficiency optimization, cost reduction, quality improvement, and effective risk management. Additionally, the IPD model emphasizes risk and benefit sharing to ensure unified collaboration among all stakeholders. Therefore, the owner shares all profits of the project with the participants. Meanwhile, in the IPD projects, the profits obtained by each member are greater than the costs invested, i.e., each member receives some net profit.*

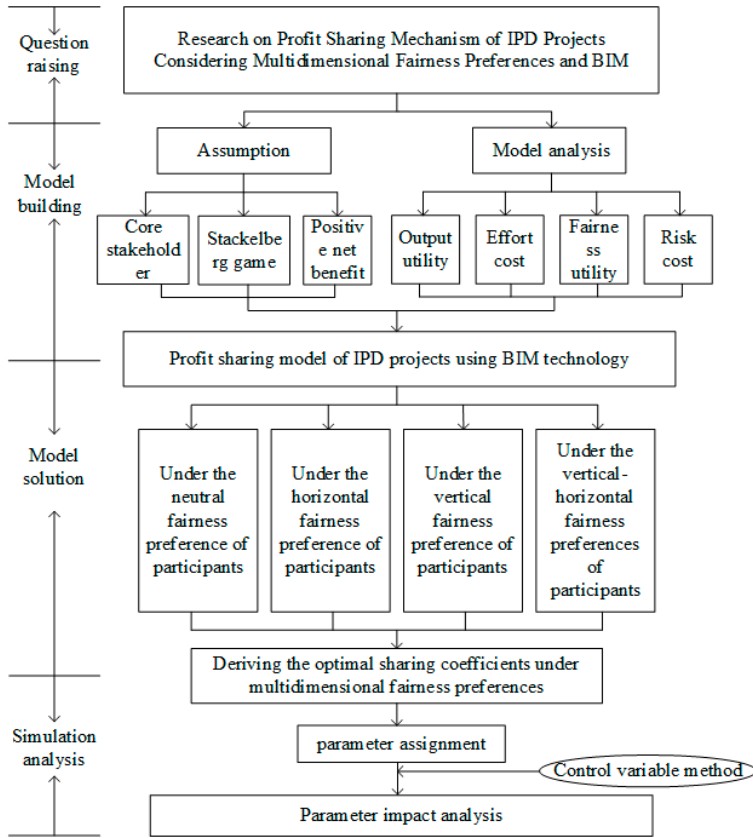

**Figure 1.** Flow chart of the research process for IPD project profit-sharing mechanism.

*2.2. Model Construction and Analysis*

In the multi-party contract of the integrated project delivery (IPD) mode, the project parties will jointly agree on the objectives and scope of the project, the roles and responsibilities of participants, risk and reward sharing, the profit-sharing mechanism, etc., to ensure the cooperation, coordination, and orderly progress of the project. One of the core design principles of the IPD project alliance contract is "shared risk, shared profit." During the process of profit sharing, it is possible to promote the stable operation of the cooperative alliance by adhering to principles such as "symmetry between profits and risks ", as well as "balance between input and return" and "equitable sharing" [46]. We list in Table 1 all the notation used throughout the paper.

**Table 1.** Notation and variable definitions.

| Notation | Definition |
|---|---|
| $I_i$ | The output utility of the participants. The measurement of the output of various behaviors such as technological contributions and shared resource input risks. |
| $a_i$ | The level of effort of participants. Namely the degree of effort in early involvement, resource investment, solidarity and collaboration, and other aspects. |
| $\pi_i$ | Effort utility coefficient of participant $i$. Refers to the utility brought by a unit of effort. |
| $\varepsilon$ | An externally random interfering normal distribution variable with a mean of zero and a variance of $\sigma^2$. |

**Table 1.** *Cont.*

| Notation | Definition |
|---|---|
| $\beta_i$ | The coefficient of profit sharing obtained by the participant $i$. |
| $C_{ai}$ | Cost of effort by the participant $i$. |
| $k_i$ | The effort cost coefficient of participant $i$. Refers to the cost incurred for each unit of effort. |
| $g$ | Owner provides fixed compensation for the participant in terms of resource consumption and other factors. |
| $S_i, S_j$ | Total revenue of Participant $i$, Participant $j$. |
| $e$ | Fairness utility. Refers to the impact of utility generated by fairness preferences. |
| $p$ | The intensity of fairness preference. Refers to the degree of perception towards fairness. |
| $U$ | The owner's actual utility, that is, the net income without considering the cost of risk. |
| $V_i$ | The participant $i$'s actual utility, that is, the net income without considering the cost of risk. |
| $C_{ri}$ | The participant $i$'s risk cost. Refers to the cost incurred when taking a series of measures to avoid risks. |
| $\rho_i$ | Participant's risk aversion coefficient. Namely the degree of risk avoidance. |
| $u$ | The deterministic equivalent utility of the owner, the final utility after considering various factors. |
| $v_i$ | The deterministic equivalent utility of the participant $i$, the final utility after considering various factors. |

### 2.2.1. Output Utility of the Participants

In IPD projects, the architect realizes the coordination and integration of multiple units and types of work based on BIM technology, and the owner and contractor communicate and exchange information using the BIM platform provided by the architect, thereby greatly promoting collaboration among all parties, reducing wastage of project resources, and increasing project profits. The diversity of the project's nature, contract agreements, and the roles and contributions of the parties lead to a variety of profit-sharing methods for the entire project. Some portions might not engage in sharing (such as proprietary interests), while certain segments could be equally profitable among all parties (such as shared profits from resources like equipment and technological platforms). Moreover, different sharing approaches might be adopted for some segments (e.g., based on technical contributions, resource inputs, goal achievements). As the extent of technological contribution, the quantity of resource input, and the degree of goal achievement are determined by participants' optimal outputs, the profits of this aspect can be measured using the participants' optimal outputs. Based on the research results of Holmstrom and Milgrom [47], the optimal output of each party involved in the project is a one-dimensional variable with a positive correlation with the level of effort. Then, in the implementation of the IPD projects, the output utility of the participants is related to their level of effort (including early participation enthusiasm, risk sharing, mutual collaboration, etc.), and is also related to their effort utility value and exogenous uncertainty. Let the output utility of participant $i$ in IPD projects be $I_i$, and then it can be expressed as

$$I_i = a_i \pi_i + \varepsilon \tag{1}$$

where $a_i(0 \leq a_i \leq 1)$ is the effort level of participant $i$; $\pi_i$ is the utility coefficient of participant $i$'s unit effort level; and $\varepsilon$ is an external random disturbance variable that follows a normal distribution $\varepsilon \sim N(0, \sigma^2)$.

In IPD projects, the output utility $I_i$ of participant $i$ is distributed between the owner and participant $i$. Between the owner and participant $i$, let the coefficient of sharing for the owner be $1 - \beta_i (0 \leq \beta_i \leq 1)$, and for participant $i$ be $\beta_i$. Then, among all members, the sharing coefficient of the owner is $1 - \sum\limits_{i=1}^{2} \beta_i$, and of participant $i$ is $\frac{\beta_i I_i}{\sum\limits_{i=1}^{2} I_i}$. Set $g$ as a fixed reward, which is a kind of physical compensation for resource consumption in the project [31]. Then, at this point, the total revenue $S_y$ for the owner and the total revenue $S_i$ for participant $i$ are expressed as

$$S_y = (1 - \beta_i)I_i - g = (a_i\pi_i + \varepsilon)(1 - \beta_i) - g \tag{2}$$

$$S_i = g + I_i\beta_i = g + (a_i\pi_i + \varepsilon)\beta_i \tag{3}$$

### 2.2.2. Cost of Effort of Participants

In the IPD projects, it is necessary for each participant to pay the corresponding effort cost to move the project forward. Although the effort cost of participants is not easily measured directly, according to the research findings of Rees [48], the effort cost is quadratically correlated with the effort cost coefficient. Then, as the effort cost of the participants can be measured by their effort level and effort cost coefficient, there is $C_{ai} = g(a_i k_i)$, where $k_i (k_i > 0)$ describes the effort cost coefficient of participant $i$. The larger the $k_i$ value is, the higher the cost of the unit effort paid by the participants, which can be obtained through evaluation or experience. Obviously, the effort cost $k_i$ of participant $i$ is proportional to its effort level $a_i$, i.e., $g(a_i k_i)$ is an increasing function of the effort level $a_i$, so $\frac{\partial C_{ai}}{\partial a_i} > 0$; moreover, the marginal cost of effort is increasing also, i.e., $\frac{\partial^2 C_{ai}}{\partial a_i^2} > 0$. Therefore, the effort cost of participant $i$ can be set to

$$C_{ai} = \frac{1}{2} k_i a_i^2 \tag{4}$$

### 2.2.3. Fairness Utility of the Participants

Fairness preference is a kind of psychological behavior of the participants about whether the sharing of the profits they have is fair. Fairness preference means that participants care not only about their own profits but also about the profits of other members, and that disparities between profits also affect their total utility [49]. According to the classical F-S model [50], if one gains less than others, one will have an additional negative utility due to jealousy, called negative jealousy utility, while if one gains more than others, one will have an additional negative utility due to guilt, called negative guilt utility. Thus, the fairness utility function $e$ of the participants can be expressed by using the F-S model

$$e = P' \max\left[ (S_i - S_j), 0 \right] - P'' \max\left[ (S_j - S_i), 0 \right] \tag{5}$$

where $P'(P' > 0)$, $P''\left(P'' > 0\right)$ are horizontal (or vertical) pride preference strength and jealousy preference strength; $S_i$, $S_j$ are the total revenues of participant $i$ and participant $j$. For ease of calculation, here make $P' = P'' = p$ in equation (7). Then the fairness utility of participant $i$ is

$$e_i = p(S_i - S_j) \tag{6}$$

### 2.2.4. Cost of Risk for the Participants

There are certain risks in the implementation of the IPD projects, including collaboration risks, technological risks, natural risks, political and legal risks, market risks, and economic risks. According to the research findings of Arrow [51], owners are usually risk-neutral and participants are risk-averse. Therefore, the participants will take corresponding measures to mitigate risks and incur risk costs.

Let the actual utility of the owner and participants be $U$ and $V_i$ respectively, which represent their net earnings without considering risk costs. Based on the total revenue $S_j$ of the owner, the total revenue $S_i$ of the participants, the effort cost $C_{ai}$ and the fairness utility $e_i$, the actual utility of the owner and participant $i$ can be calculated as

$$U = I_i(1 - \beta_i) - g \tag{7}$$

$$V_i = S_i + e_i - C_{ai} \tag{8}$$

According to the research of Fu and Zhu [52], the cost of risk $C_{ri}$ for participant $i$ can be described as $C_{ri} = \frac{1}{2} \rho_i D(V_i)$, where $\rho_i(\rho_i \geq 0)$ is a measure of risk aversion (avoidance) of participant $i$, i.e., the risk aversion coefficient. A larger value of $\rho_i$ indicates a higher degree of risk aversion. $D(V_i)$ represents the variance of the actual utility $V_i$ for participant

*i*. From Equations (3), (4) and (8), the specific expression of the participant' risk cost $C_{ri}$ can be obtained as

$$
\begin{aligned}
C_{ri} &= \tfrac{1}{2}\rho_i D(V_i) \\
&= \tfrac{1}{2}\rho_i D(S_i + e_i - C_{ai}) \\
&= \tfrac{1}{2}\rho_i D[g + (a_i\pi_i + \varepsilon)\beta_i - \tfrac{1}{2}k_i a_i{}^2 + e_i] \\
&= \tfrac{1}{2}\rho_i[\sigma^2\beta_i^2 + D(e_i)]
\end{aligned}
\tag{9}
$$

Here, $\sigma$ represents the standard deviation of the normal distribution that the external random disturbance variable $\varepsilon$ follows, $\varepsilon \sim N(0, \sigma^2)$.

From the above, the deterministic equivalent utility for the owner and participants in IPD projects are expressed as

$$
u = a_i\pi_i(1 - \beta_i) - g
\tag{10}
$$

$$
v_i = S_i - C_{ai} - C_{ri} = g + a_i\pi_i\beta_i - \frac{1}{2}k_i a_i{}^2 - \frac{1}{2}\rho_i D(S_i + e_i - C_{ai})
\tag{11}
$$

### 2.3. Model Solution

In the owner-led Stackelberg game, the owner takes the lead in deciding about the profit-sharing coefficient. Subsequently, the participants adjust their individual effort levels based on these decisions to achieve the optimal response. Therefore, the total revenue function of the owner is the objective function of the model, and the optimal response function $a(\beta)$ of the participants is the constraint. Meanwhile, combining the multidimensional fairness preferences of the participants using the inverse solution method, the optimal profit-sharing coefficient and effort level of the model under each fairness preference dimension can be obtained.

#### 2.3.1. Under Fairness Preference Neutrality

When participant *i* has a neutral fairness preference, its actual utility $V_{wi}$ is

$$
V_{wi} = S_i - C_{ai} = g + (a_i\pi_i + \varepsilon)\beta_i - \frac{1}{2}k_i a_i{}^2
\tag{12}
$$

Therefore, it is obtained that $E(V_{wi}) = E(S_i - C_{ai}) = g + a_i\pi_i\beta_i - \frac{1}{2}k_i a_i{}^2$, $D(V_{wi}) = D(S_i - C_{ai}) = \sigma_i^2\beta_i{}^2$.

Then, the owner's deterministic equivalent utility $u_w$ and participant *i*'s deterministic equivalent utility $v_{wi}$ are

$$
u_w = a_i\pi_i(1 - \beta_i) - g
\tag{13}
$$

$$
v_{wi} = g + a_i\pi_i\beta_i - \frac{1}{2}k_i a_i{}^2 - \frac{1}{2}\rho_i\sigma^2\beta_i{}^2
\tag{14}
$$

In IPD projects, the owner should ensure that each participant's expected utility is not lower than what they would have obtained if they had not participated in the project. Let the profit that the participate *i* would still receive if not participating in the project be $W_i$, then:

$$
g + a_i\pi_i\beta_i - \frac{1}{2}k_i a_i{}^2 - \frac{1}{2}\rho_i\sigma^2\beta_i{}^2 \geq W_i
\tag{15}
$$

After that, the participants make their own choices of the effort level according to the owner's decision result, and the optimal response function $a_i(\beta_i)$ is

$$
\begin{aligned}
\max(v_{wi}) &= \max\left(g + a_i\pi_i\beta_i - \tfrac{1}{2}k_i a_i{}^2 - \tfrac{1}{2}\rho_i\sigma^2\beta_i{}^2\right) \\
a_i &= \operatorname{argmax}(v_{wi}) = \tfrac{\pi_i\beta_i}{k_i}
\end{aligned}
\tag{16}
$$

In summary, when the participants are neutral in fairness preference, the profit-sharing model of IPD projects is as follows:

$$\max(u_w) = a_i\pi_i(1 - \beta_i) - g$$
$$s.t.\begin{cases} g + a_i\pi_i\beta_i - \frac{1}{2}k_ia_i{}^2 - \frac{1}{2}\rho_i\sigma^2\beta_i{}^2 \geq W_i \\ a_i = \text{argmax}(v_{wi}) = \frac{\pi_i\beta_i}{k_i} \end{cases} \tag{17}$$

By substituting the participation constraint into the objective function and deriving the sharing coefficient, the optimal profit-sharing coefficient $\beta_i$ and the optimal effort level $a_i$ for participant $i$ can be obtained from the first-order optimality condition as

$$\beta_i = \frac{\pi^2{}_i}{\pi_i{}^2 + k_i\rho_i\sigma^2}, a_i = \frac{\pi_i\beta_i}{k_i} \tag{18}$$

### 2.3.2. Under Horizontal Fairness Preference

The horizontal fairness preference means that participants are concerned not only about their own profits but also about those of the other participants, and the disparity between these profits also impacts their own total utility. According to Equation (7), when the horizontal fairness intensity of participant $i$ is $p_{hi}$, the horizontal fairness preference utility $e_{hi}$ is as follows:

$$e_{hi} = p_{hi}(S_i - S_j) = p_{hi}(a_i\pi_i - a_j\pi_j) \tag{19}$$

In this case, the actual utility $V_{hi}$ of participant $i$ is

$$V_{hi} = g + (a_i\pi_i + \varepsilon)\beta_i - \frac{1}{2}k_ia_i{}^2 + p_{hi}(a_i\pi_i - a_j\pi_j)\beta_i \tag{20}$$

So, $E(V_{hi}) = S_i - C_{ai} + e_{hi} = g + (a_i\pi_i + \varepsilon)\beta_i - \frac{1}{2}k_ia_i{}^2 + p_{hi}(a_i\pi_i - a_j\pi_j)\beta_i D(V_{hi}) = \sigma^2\beta^2$. The deterministic equivalent utility $u_h$ of the owner and the deterministic equivalent utility $v_{hi}$ of participant $i$ can be obtained as follows:

$$u_h = I_i(1 - \beta_i) - g = a_i\pi_i(1 - \beta_i) - g \tag{21}$$

$$\begin{aligned} v_{hi} = S_i - C_{ai} - C_{ri} + e_{zi} = g + a_i\pi_i\beta_i - \frac{1}{2}k_ia_i{}^2 \\ - \frac{1}{2}\rho_i\sigma^2\beta_i{}^2 + p_{hi}(a_i\pi_i - a_j\pi_j)\beta_i \end{aligned} \tag{22}$$

Therefore, when participants have a horizontal fairness preference, the profit-sharing model of IPD projects is as follows:

$$\max u_h = a_i\pi_i(1 - \beta) - g$$
$$s.t.\begin{cases} g + a_i\pi_i\beta_i - \frac{1}{2}k_ia_i{}^2 - \frac{1}{2}\rho_i\sigma^2\beta_i{}^2 \geq W_i \\ \max\left(g + a_i\pi_i\beta_i - \frac{1}{2}k_ia_i{}^2 - \frac{1}{2}\rho_i\sigma^2\beta_i{}^2 + p_{hi}(a_i\pi_i - a_j\pi_j)\beta_i\right) \end{cases} \tag{23}$$

By substituting the participation constraint into the objective function and deriving the sharing coefficient, the optimal profit-sharing coefficient $\beta_{hi}$ and the optimal effort level $a_{hi}$ can be obtained from the first-order optimality condition as

$$\beta_{hi} = \frac{(1 + p_{hi})\pi_i{}^2}{(1 + p_{hi})^2\pi_i{}^2 + k_i\rho_i\sigma^2}, a_{hi} = \frac{(1 + p_{hi})\pi_i\beta_{hi}}{k} \tag{24}$$

### 2.3.3. Under Vertical Fairness Preference

The vertical fairness preference of the participants means that the participants will compare their profits with the owner, and the difference in these profits will also affect

their total utility. Similar to the horizontal fairness preference, the vertical pride preference and vertical envy preference are considered together. Let $p_{zi}$ denote the intensity of the vertical fairness preference; then the utility $e_{zi}$ brought by the vertical fairness preference of participant $i$ is as follows:

$$
\begin{aligned}
e_{zi} = p_{zi}(S_i - S_y) &= p_{zi}[g + (a_i\pi_i + \varepsilon)\beta_i + g - (a_i\pi_i + \varepsilon)(1 - \beta_i)] \\
&= p_{zi}[2g + (2\beta_i - 1)(a_i\pi_i + \varepsilon)]
\end{aligned}
\tag{25}
$$

The actual utility of participant $i$ is obtained as

$$
V_{zi} = S_i - C_{ai} + e_{zi} = g + (a_i\pi_i + \varepsilon)\beta_i - \frac{1}{2}k_i a_i^2 + p_{zi}[2g + (2\beta_i - 1)(a_i\pi_i + \varepsilon)]
\tag{26}
$$

At this time, the deterministic equivalent utilities $u_z$ and $v_{zi}$ of the owner and participant $i$ are as follows:

$$
u_z = I(1 - \beta_i) - g = a_i\pi_i(1 - \beta_i) - g
\tag{27}
$$

$$
\begin{aligned}
v_{zi} = S_i - C_{ai} - C_{ri} + e_{zi} &= g + a_i\pi_i\beta_i - \tfrac{1}{2}k_i a_i^2 \\
&- \tfrac{1}{2}\rho_i\left(\beta_i^2 + p_{zi}^2(2\beta_i - 1)^2\right)\sigma^2 + p_{zi}[2g + (2\beta_i - 1)(a_i\pi_i + \varepsilon)]
\end{aligned}
\tag{28}
$$

Therefore, when participants have a vertical fairness preference, the profit-sharing model of IPD projects is as follows:

$$
\begin{aligned}
&\max u_z = a_i\pi_i\eta B(1 - \beta) - g \\
&s.t. \begin{cases} g + a_i\pi_i\beta_i - \tfrac{1}{2}k_i a_i^2 - \tfrac{1}{2}\rho_i\left(\beta_i^2 + p_{zi}^2(2\beta_i - 1)^2\right)\sigma^2 \geq W_i \\ \max\left( \begin{array}{l} g + a_i\pi_i\beta_i - \tfrac{1}{2}k_i a_i^2 - \tfrac{1}{2}\rho_i\left(\beta^2 + p_{zi}^2(2\beta_i - 1)^2\right)\sigma^2 \\ + p_{zi}[(2g + (2\beta_i - 1)a_i\pi_i)] \end{array} \right) \end{cases}
\end{aligned}
\tag{29}
$$

By substituting the constraint into the objective function and taking the derivative of the profit-sharing coefficient, the optimal profit-sharing coefficient $\beta_{zi}$ and effort level $a_{zi}$ can be obtained by the first-order optimization condition as

$$
\beta_{zi} = \frac{\pi_i^2(1 + 3p_{zi}'' + 2p_{zi}''^2) + 2k_i p_{zi}^2\rho_i\sigma^2}{(4p_{zi}^2 + 4p_{zi} + 1)\pi_i^2 + k\rho_i\sigma^2(4p_{zi}^2 + 1)}, a_{zi} = \frac{[p_{zi}(2\beta_i - 1) + \beta_{zi}]\pi_i}{k_i}
\tag{30}
$$

### 2.3.4. Under Vertical–Horizontal Fairness Preference

The vertical–horizontal fairness preference of the participants means that they not only consider their own profits but also take into account the profits of the owner and other participants. Additionally, the disparities in these profits also influence the total utility of the participants themselves. The impact of the vertical–horizontal fairness preference on participants' total utility is equivalent to the combined effects of the individual fairness preferences, both vertical and horizontal, respectively. Let $p_{qi}$ represent the intensity of the vertical–horizontal fairness preference of participant $i$; then the vertical–horizontal fairness preference utility $e_{qi}$ is

$$
e_{qi} = p_{qi}[(S_i - S_y) + (S_i - S_j)] = p_{qi}\begin{bmatrix} (2g + (2\beta_i - 1)(a_i\pi_i + \varepsilon)) \\ + (a_i\pi_i - a_j\pi_j)\beta_i \end{bmatrix}
\tag{31}
$$

The actual utility of participant $i$ is obtained as follows:

$$
\begin{aligned}
V_{qi} = S_i - C_{ai} + e_{qi} &= g + (a_i\pi_i + \varepsilon)\beta_i - \tfrac{1}{2}k_i a_i^2 \\
&+ p_{qi}\begin{bmatrix} (2g + (2\beta_i - 1)(a_i\pi_i + \varepsilon)) \\ + (a_i\pi_i - a_j\pi_j)\beta_i \end{bmatrix}
\end{aligned}
\tag{32}
$$

At this time, the deterministic equivalent utility $u_q$ and $v_{qi}$ of the owner and participant $i$ are, respectively,

$$u_q = a_i \pi_i (1 - \beta_i) - g \tag{33}$$

$$v_{qi} = S_i - C_{ai} - C_{ri} + e_{qi} = g + a_i \pi_i \beta_i - \tfrac{1}{2} k_i a_i^2$$
$$- \tfrac{1}{2} \rho_i \left( \beta_i^2 + p_{qi}^2 (2\beta_i - 1)^2 \right) \sigma^2 + p_{qi} \left[ \begin{array}{c} (2g + (2\beta_i - 1)a_i \pi_i) \\ + (a_i \pi_i - a_j \pi_j) \beta_i \end{array} \right] \tag{34}$$

Therefore, when participants have a vertical–horizontal fairness preference, the profit-sharing model of IPD projects is as follows:

$$\max u_q = a_i \pi_i (1 - \beta_i) - g$$
$$s.t. \begin{cases} g + a_i \pi_i \beta_i - \tfrac{1}{2} k_i a_i^2 - \tfrac{1}{2} \rho_i \left( \beta_i^2 + p_{qi}^2 (2\beta_i - 1)^2 \right) \sigma^2 \geq W_i \\ \max \left( \begin{array}{c} g + a_i \pi_i \beta_i - \tfrac{1}{2} k_i a_i^2 - \tfrac{1}{2} \rho_i \left( \beta_i^2 + p_{qi}^2 (2\beta_i - 1)^2 \right) \sigma^2 \\ + p_{qi} \left[ \begin{array}{c} (2g + (2\beta_i - 1)a_i \pi_i) \\ + (a_i \pi_i - a_j \pi_j) \beta_i \end{array} \right] \end{array} \right) \end{cases} \tag{35}$$

By substituting the participation constraint into the objective function and taking the derivative of the profit-sharing coefficient, the optimal profit-sharing coefficient $\beta_{qi}$ and effort level $a_{qi}$ can be obtained from the first-order optimization condition as

$$\beta_{qi} = \frac{\left(1 + 4p_{qi} + 3p_{qi}^2\right)\pi^2 + 2k\rho\sigma^2 p_{qi}^2}{\left(1 + 3p_{qi}\right)^2 \pi^2 + k\rho\sigma^2 \left(1 + 4p_{qi}^2\right)}, a_{qi} = \frac{\beta_{qi}\left(1 + 3p_{qi}\right)\pi_i - p_{qi}\pi_i}{k_i} \tag{36}$$

### 3. Simulation Analysis

In an IPD project, the collaborating entities collectively form a cooperative alliance aimed at fulfilling the owner's requisites and ensuring the triumphant delivery of project functionality or value. Nevertheless, as distinct economic entities, these stakeholders must also contemplate the maximization of their individual aggregate returns. According to the above model, the total income of the participants is closely linked to their output utility and the coefficient of the shared benefits acquired. Within an IPD project, the output utility of the participants directly influences the team's performance as a whole and the cumulative project revenue, and it may even affect the overall success of the project. The profit-sharing coefficient encompasses various dimensions, including balancing contributions from different parties, ensuring fairness, incentivizing participation, and distributing risks. It stands as a pivotal element in the establishment of a mechanism for mutual benefit sharing. Therefore, the output utility, profit-sharing coefficient, and total revenue of the participants can be regarded as the dependent variables for the study of the sharing mechanism. Through simulation, the influence of other parameters on them can be undertaken. To maximize the revenue of the project alliance, all participants must make essential compromises and contributions. As each participant represents a heterogeneous enterprise with distinct roles and core competitive advantages, their unit effort costs and values may vary. Consequently, their contributions within the project differ due to these discrepancies. Furthermore, profit sharing follows the principle of "High risk, High return". Due to varying risk aversion coefficients among different participating entities, the extent to which risks are avoided differs. Simultaneously, the fairness preference, as a form of perceptual behavior regarding equitable profit distribution, influences the effort level of participants within the IPD project and plays a crucial role in the project's success.

To explore the profit-sharing mechanism of IPD projects, this study uses simulation with controlled variables to analyze the impact of key parameters, such as participants' effort cost coefficient $k$, effort utility value $\pi$, and fairness preference intensity $P$, on their output utility $I$, profit sharing coefficient $\beta$, and total revenue $S$.

The case studies of the IPD mode collected by AIA [45] are combined with the parameter settings for the risk aversion coefficient and risk standard deviation in the research conducted by Guo et al. [31], as well as the parameter settings for the effort cost coefficient and effort utility value in the study by Onur [53]. For an IPD project, the main members include the owner, architect, and contractor, and the specific parameter values are shown in Table 2.

**Table 2.** Relevant parameters.

| Parameters | Values | Parameters | Values | Parameters | Values |
|---|---|---|---|---|---|
| $k_1$ (Owner's effort cost coefficient /ten thousand yuan) | 2.2 | $k_2$ (Architect's effort cost coefficient /ten thousand yuan) | 2.5 | $k_3$ (Contractor's effort cost coefficient /ten thousand yuan) | 1.8 |
| $\pi_1$ (Owner's effort utility coefficient/ten thousand yuan) | 2.5 | $\pi_2$ (Architect's effort utility coefficient/ten thousand yuan) | 3 | $\pi_3$ (Contractor's effort utility coefficient/ten thousand yuan) | 2 |
| $\rho_1$ (The risk aversion coefficient of the owner) | 0.5 | $\rho_2$ (The risk aversion coefficient of architect) | 3 | $\rho_3$ (The risk aversion coefficient of contractor) | 2.8 |
| $\sigma_1$ (Risk-averse standard deviation for owner) | 0.5 | $\sigma_2$ (Risk-averse standard deviation of architect) | 1.2 | $\sigma_3$ (Risk-averse standard deviation of contractor) | 1.2 |
| $p_{q2}$ (Vertical–horizontal fairness preference intensity of architect) | 0.5 | $p_{q3}$ (Vertical–horizontal fairness preference intensity of contractor) | 0.5 | $p_{h3}$ (Horizontal fairness preference intensity of contractor) | 0.5 |
| $p_{h2}$ (Horizontal fairness preference intensity of architect) | 0.5 | $p_{z2}$ (Vertical fairness preference intensity of architect) | 0.5 | $p_{z3}$ (Vertical fairness preference intensity of contractor) | 0.5 |

The output utility $I$, profit-sharing coefficient $\beta$, and total revenue $S$ are calculated, respectively, and they are shown in Table 3.

**Table 3.** Calculated values.

| | Owner | | Architect | | Contractor | |
|---|---|---|---|---|---|---|
| | | Under the fairness preference neutrality of the participants | | | | |
| | | | $I_{w2}$ | 1.6362 | $I_{w3}$ | 0.7896 |
| $\beta_{w1}$ | 0.5777 | | $\beta_{w2}$ | 0.3066 | $\beta_{w3}$ | 0.1157 |
| $S_{w1}$ | 1.4014 | | $S_{w2}$ | 0.7437 | $S_{w3}$ | 0.2807 |
| | | Under the horizontal fairness preference of the participants | | | | |
| | | | $I_{h2}$ | 2.3478 | $I_{h3}$ | 1.2304 |
| $\beta_{h1}$ | 0.5878 | | $\beta_{h2}$ | 0.2853 | $\beta_{h3}$ | 0.1269 |
| $S_{h1}$ | 2.1032 | | $S_{h2}$ | 1.0209 | $S_{h3}$ | 0.4541 |
| | | Under the vertical fairness preference of the participants | | | | |
| | | | $I_{z2}$ | 2.16 | $I_{z3}$ | 0.9265 |
| $\beta_{z1}$ | 0.5190 | | $\beta_{z2}$ | 0.3850 | $\beta_{z3}$ | 0.0960 |
| $S_{z1}$ | 1.6019 | | $S_{z2}$ | 1.1883 | $S_{z3}$ | 0.2963 |
| | | Under the vertical and horizontal fairness preference of the participants | | | | |
| | | | $I_{q2}$ | 2.7261 | $I_{q3}$ | 1.5078 |
| $\beta_{q1}$ | 0.5083 | | $\beta_{q2}$ | 0.3238 | $\beta_{q3}$ | 0.1679 |
| $S_{q1}$ | 2.1521 | | $S_{q2}$ | 1.3710 | $S_{q3}$ | 0.7108 |

According to Table 2, under the multidimensional fairness preference of participants, their corresponding output utility $I$, profit-sharing coefficient $\beta$, and total revenue $S$ are different. This is due to the different sources of fairness perception of fairness preferences in each dimension, which brings a different fairness utility function $e$ to the participants. As a result, their actual utility function is different, and the optimal sharing coefficient obtained by solving is also different.

In IPD projects, all participants form an intricate and closely connected interest alliance. However, when the sharing of profits obtained by the participants does not meet their expectation, their enthusiasm to participate in the project will be affected, and it will ultimately lead to the collapse of the entire alliance group. To further analyze the impact of various parameters on participants' output utility $I$, profit-sharing coefficient $\beta$, and total revenue $S$ under multidimensional fairness preferences, the parameter values of the architect are taken as an example, and the simulation is carried out on the basis of control variables. The results are shown in Figures 2–10.

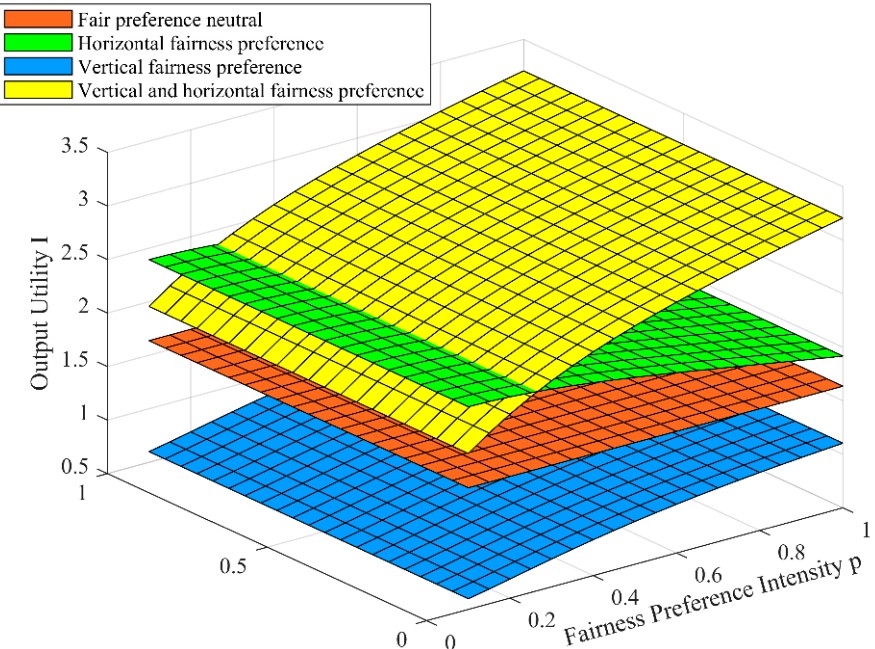

**Figure 2.** The effect of $P$ on $I$.

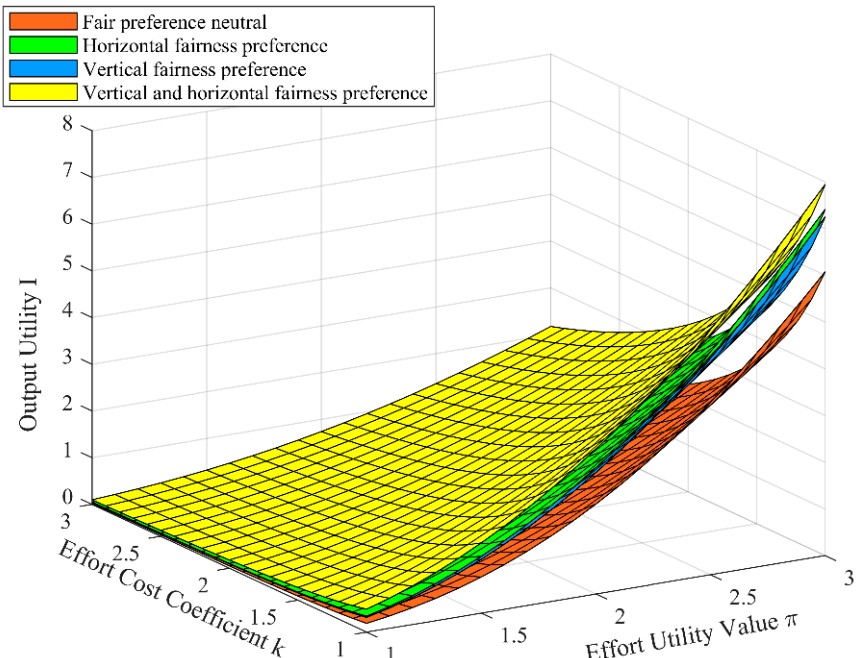

**Figure 3.** The effect of $k$ and $\pi$ on $I$.

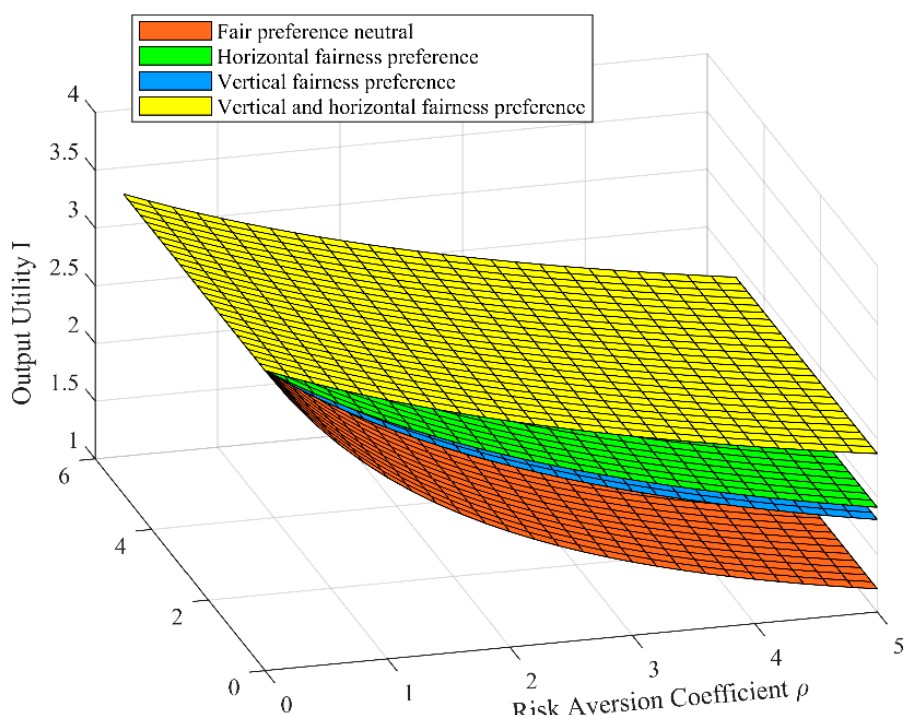

**Figure 4.** The effect of $\rho$ on $I$.

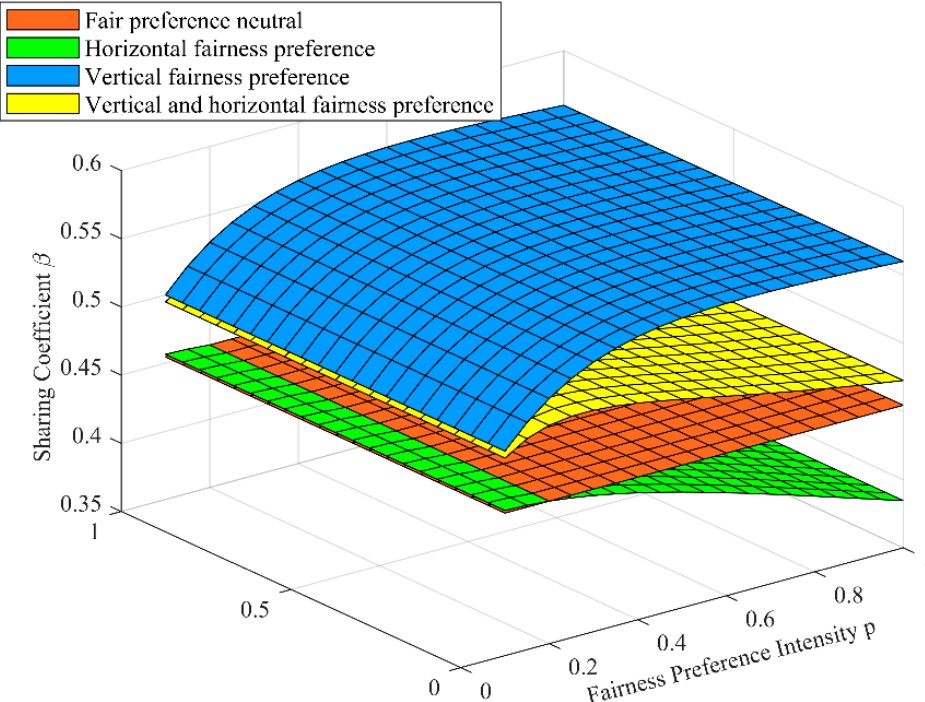

**Figure 5.** The effect of $P$ on $\beta$.

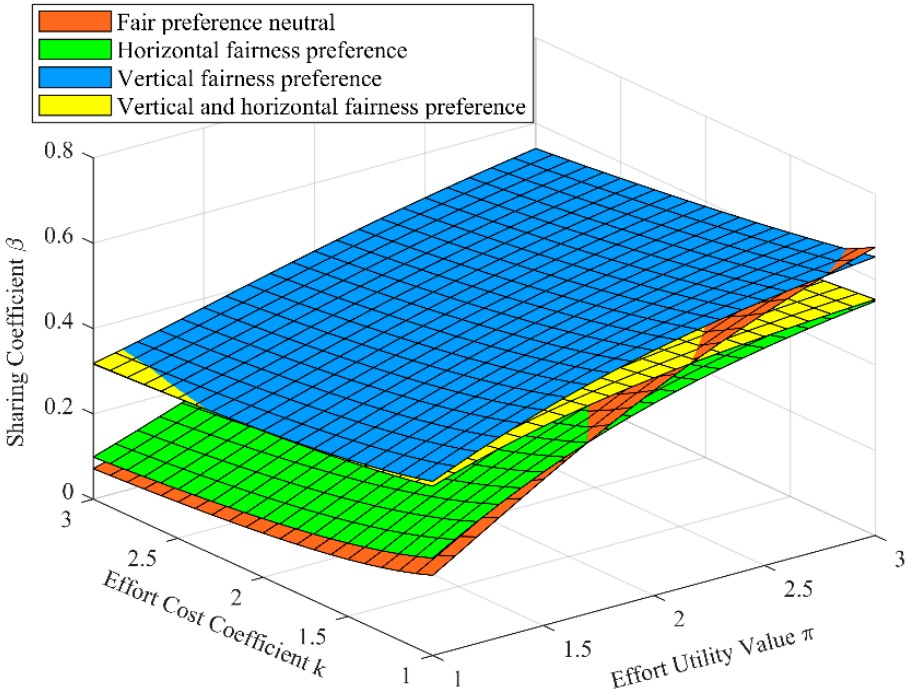

**Figure 6.** The effect of $k$ and $\pi$ on $\beta$.

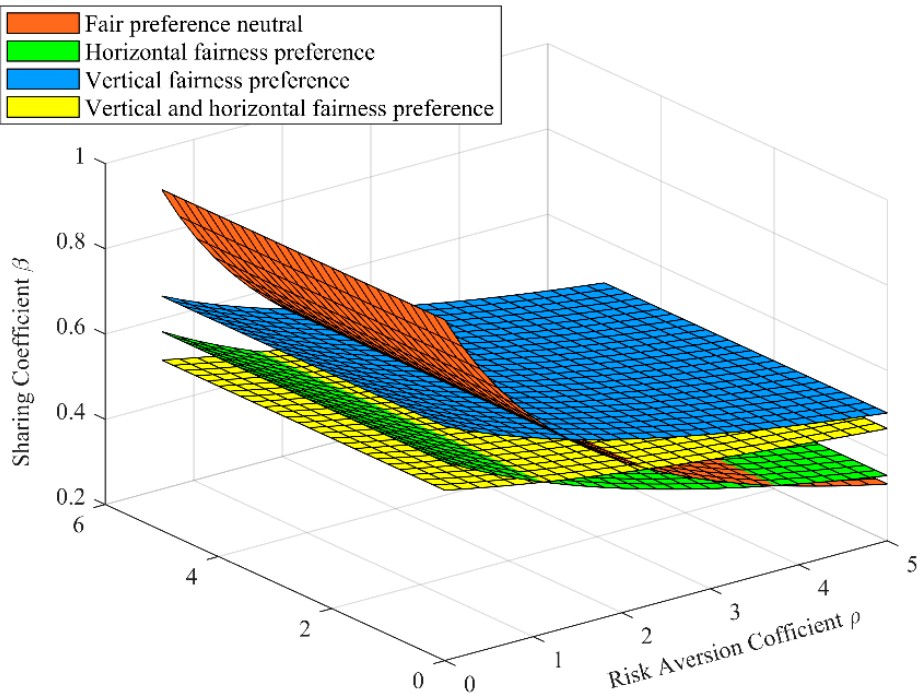

**Figure 7.** The effect of $\rho$ on $\beta$.

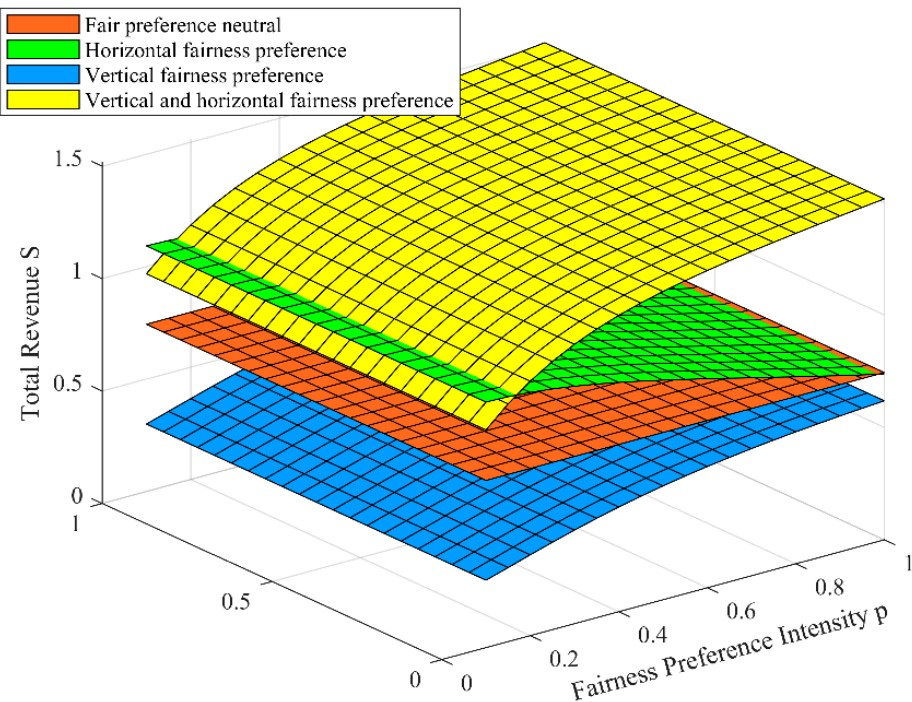

**Figure 8.** The effect of *P* on *S*.

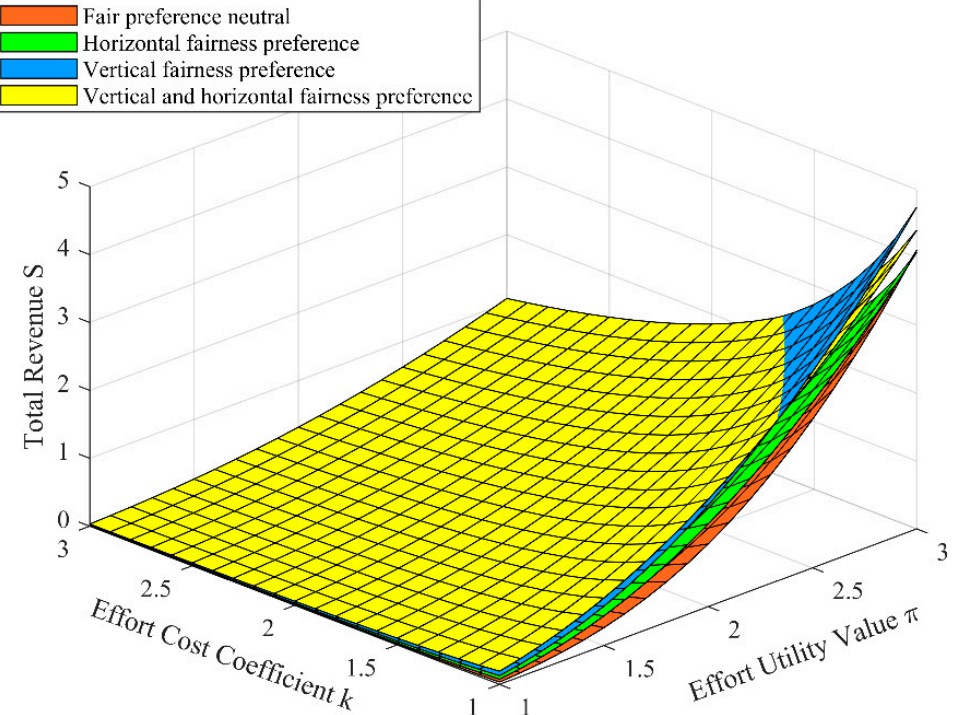

**Figure 9.** The effect of *k* and $\pi$ on *S*.

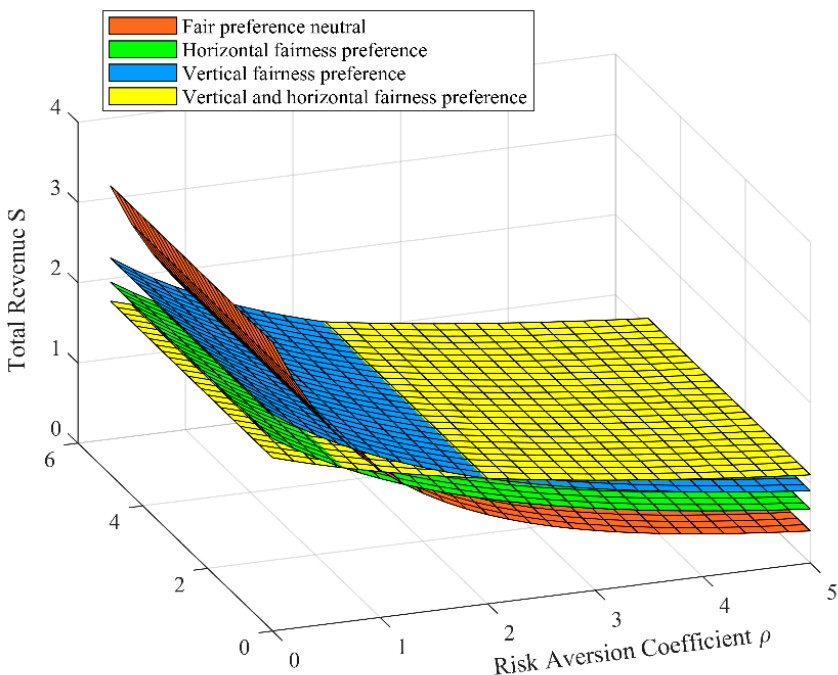

**Figure 10.** The effect of $\rho$ on $S$.

*3.1. The Effect on the Output Utility of the Participants*

3.1.1. The Effect of Participants' Fairness Preference Intensity on Their Output Utility

The effect of participants' fairness preference intensity on their output utility is shown in Figure 2. It can be seen from Figure 2 that compared with neutral fairness preferences, participants' horizontal fairness preferences and vertical–horizontal fairness preferences both enhance output utility, while vertical fairness preferences reduce it. Furthermore, with the increase in the intensity of participants' vertical and vertical–horizontal fairness preferences, their output utility increases. Conversely, as the intensity of horizontal fairness preferences strengthen, participants' output utility decreases.

It can be seen that when the participants have a horizontal fairness preference, there will be comparison and competition among the participants. In order to obtain higher income, the participants will actively improve their effort level independently. Therefore, compared with neutral fairness preferences, horizontal fairness preferences can improve the output utility. However, with the increase of the horizontal fairness preference intensity, the competition among the participants becomes increasingly fierce, which is not conducive to the team cooperation of IPD projects, resulting in the decline of their output utility. When the participants have a vertical fairness preference, the income gap between the participants and the owner will make them produce unfair psychology and affect their effort level. Therefore, compared with the neutral fairness preference, the vertical fairness preference of the participants reduces the output utility. Meanwhile, with the increase of vertical fairness preference intensity, the owner will take incentive measures to improve the effort level of the participants so as to promote the increase of the output utility of the participants with the increase of vertical fairness preference intensity.

3.1.2. The Effect of Participants' Effort Utility Value and Cost Coefficient on Output Utility

The effect of participants' effort utility value and cost coefficient on output utility is shown in Figure 3. It can be seen from Figure 3 that under the multidimensional fairness preference, participants' output utility increases with the increase of the effort utility value and decreases with the increase of the effort cost coefficient. Moreover, the output utility is more sensitive to changes in the value of effort utility. In addition, under the four dimensions of fairness preference, when all parameters are fixed, the participants' output utility is the highest under the vertical–horizontal fairness preference, followed

by the horizontal fairness preference, the vertical fairness preference, and the neutral fairness preference.

It can be seen that in the IPD projects, as the utility value of the participants' effort increases and the cost coefficient decreases, the level of effort that the participants are willing to exert will increase, so their output utility will increase. In addition, when the participants have a horizontal fairness preference, the competition among the participants will make them actively improve their level of effort, thereby increasing their output utility. When the participants have a vertical horizontal fairness preference, the interest gap between the participants and the owner will affect the enthusiasm of the participants' effort. Therefore, the owner will take incentive measures to mobilize the participants' effort level so as to increase the output utility of the participants. The vertical–horizontal fairness preference combines both vertical and horizontal fairness preferences. Therefore, when the participants have a vertical–horizontal fairness preference, their output utility reaches its maximum level.

### 3.1.3. The Effect of Participants' Risk Aversion Coefficient on Their Output Utility

The effect of participants' risk aversion coefficient on their output utility is shown in Figure 4. It can be seen from Figure 4 that under the fairness preference of all dimensions, participants' output utility decreases with the increase of the risk aversion coefficient. It can be seen that in IPD projects, when participants' risk aversion coefficient increases, their aversion to risk intensifies, leading to a decrease in the risks they are willing to bear. This decrease in willingness to take on risks corresponds to a decrease in the contributions made to the IPD project, ultimately resulting in a decrease in output utility.

### 3.2. The Effect on the Profit-Sharing Coefficient of the Participants

#### 3.2.1. The Effect of Participants' Fairness Preference Intensity on Their Profit-Sharing Coefficient

The effect of participants' fairness preference intensity on their profit-sharing coefficient is shown in Figure 5. It can be seen from Figure 5 that compared with the neutral fairness preference, participants' vertical fairness preference can enhance the profit-sharing coefficient they receive, and with the increase of the intensity, the profit-sharing coefficient will also increase. The horizontal fairness preference can reduce the profit-sharing coefficient, and as the preference intensity increases, the coefficient decreases. And the vertical–horizontal fairness preference can enhance the profit-sharing coefficient, and as the intensity of this preference increases, the coefficient declines.

It can be seen that in the IPD projects, when the intensity of the participants' vertical fairness preference increases, the income gap between the participants and owner will make the participants produce unfair psychology, which will reduce their effort level. Therefore, the owner will increase the profit-sharing coefficient to improve their effort. When the intensity of the participants' horizontal fairness preference increases, the competition between the participants becomes increasingly fierce, which is not conducive to the team collaboration of IPD projects. In order to reconcile the contradictions, the owner will reduce the profit-sharing coefficient of the participants. The vertical–horizontal fairness preference has the characteristics of both vertical and horizontal fairness preferences, so its effect on the profit-sharing coefficient is a combination of the effect of both the vertical and horizontal fairness preferences.

#### 3.2.2. The Effect of Participants' Effort Utility Value and Cost Coefficient on Profit-Sharing Coefficient

The effect of participants' effort utility value and cost coefficient on the sharing coefficient is shown in Figure 6. It can be seen from Figure 6 that under each fairness preference dimension, participants' sharing coefficient increases with the rising effort utility value and decreases with the increasing effort cost coefficient. Meanwhile, the sharing coefficient is more sensitive to effort utility value. In addition, in the initial state ($k = 2.5, \pi = 3$), a vertical fairness preference allows participants to achieve a high sharing coefficient. With the

increase of the effort utility value and the decrease of the effort cost coefficient, the neutral fairness preference becomes capable of yielding a high sharing coefficient for participants.

It can be seen that in IPD projects, the greater the utility value of the participants' effort and the lower the effort cost coefficient, the more contribution per unit effort will be brought, and the effort level that participants are willing to pay will also be improved. Therefore, according to the sharing principle of "More pay for more work", the profit-sharing coefficient that participants will obtain should also be increased. Secondly, since the vertical fairness preference of the participants has a positive effect on the improvement of the profit-sharing coefficient, the horizontal fairness preference has a negative effect on it. Therefore, participants with a vertical fairness preference can obtain a higher profit-sharing coefficient. At the same time, the improvement of the participants' effort utility value and the reduction of the effort cost coefficient inherently facilitate an increase in the sharing coefficient. Therefore, when participants' effort utility value increases and the effort cost coefficient decreases to a certain extent, a substantial sharing coefficient can be achieved even under a neutral fairness preference.

### 3.2.3. The Effect of Participants' Risk Aversion Coefficient on Their Profit-Sharing Coefficient

The effect of participants' risk aversion coefficient on their sharing coefficient is shown in Figure 7. It can be seen from Figure 7 that, under each fairness preference dimension, the sharing coefficient obtained by participants decreases as the risk aversion coefficient increases. Furthermore, in the initial state ($\rho = 3$), the participants with vertical fairness preferences can obtain a high sharing coefficient. As the risk aversion coefficient diminishes, the participants with neutral fairness preferences can obtain a high sharing coefficient.

It can be seen that in IPD projects, the increase of participants' risk aversion coefficient means that their degree of risk aversion increases, and the risk they are willing to take decreases accordingly. Therefore, according to the profit-sharing principle of "High risk and high return", the sharing coefficient obtained by participants should also be reduced. Furthermore, participants' vertical fairness preference can improve their sharing coefficient, while the horizontal fairness preference will reduce it. Therefore, in the initial state ($\rho = 3$), participants can obtain a high profit-sharing coefficient with the vertical fairness preference. At the same time, the reduction of the risk aversion coefficient can promote the improvement of the profit-sharing coefficient itself. Therefore, when the risk aversion coefficient drops to a certain value, the participants with neutral fairness preferences can obtain a high sharing coefficient.

### 3.3. The Effect on the Total Revenue of the Participants
### 3.3.1. The Effect of Participants' Fairness Preference Intensity on Their Total Revenue

The effect of participants' fairness preference intensity on their total revenue is shown in Figure 8. It can be seen from Figure 8 that compared to the neutral fairness preference, participants' horizontal fairness preference can increase the total revenue, while with the increase of the intensity, the total revenue will decrease. The vertical fairness preference can reduce the total revenue, while with the increase of the intensity, the total revenue will increase. The vertical–horizontal fairness preference can improve the total revenue, and with the increase of the fairness preference intensity, the total revenue will increase.

It can be seen that the improvement of the participants' output utility caused by their horizontal fairness preference can promote the increase of the total revenue. However, with the increase of the intensity, the owner will reduce the profit-sharing coefficient of the participants, resulting in a decline in their total income. The reduction of the participants' output utility caused by their vertical fairness preference can lead to a decline in total revenue, as the increase of the intensity and the increase of the profit-sharing coefficient of the owner to the participants contributes to an increase in participants' total revenue.

### 3.3.2. The Effect of Participants' Effort Utility Value and Cost Coefficient on Their Total Revenue

The effect of participants' effort utility value and cost coefficient on their total revenue is shown in Figure 9. It can be seen from Figure 9 that under each fairness preference dimension, the participants' total revenue will increase with the increase of the effort utility value and will decrease with the increase of the effort cost coefficient, and the total revenue is more sensitive to the effort utility value. Furthermore, in the initial state ($k = 2.5, \pi = 3$), participants with vertical–horizontal fairness preferences can obtain high total revenues. With the increase of effort utility value and the decrease of the effort cost coefficient, participants with vertical fairness preferences can obtain high total revenues.

It can be seen that in IPD projects, as participants' effort utility value increases and the cost coefficient decreases, their output utility and profit-sharing coefficient will increase correspondingly, thus promoting the increase of their total revenue. Secondly, in the initial state ($k = 2.5, \pi = 3$), when participants have a vertical–horizontal fairness preference, they will obtain a high output utility, so they will also obtain a high total revenue at this time. With the increase of the effort utility value and the decrease of the effort cost coefficient, the increase of the vertical fairness preference on the profit-sharing coefficient makes the total revenue of the participants the highest under the vertical fairness preference.

### 3.3.3. The Effect of Participants' Risk Aversion Coefficient on Their Total Revenue

The effect of participants' risk aversion coefficient on their total revenue is shown in Figure 10. It can be seen from Figure 10 that under each dimension of fairness preference, participants' total revenue experiences a decline as the risk aversion coefficient increases. Furthermore, in the initial state ($\rho = 3$), the participants can obtain high total revenues under vertical–horizontal fairness preferences. With the decrease of the risk aversion coefficient, the participants first obtain high total revenues under vertical fairness preferences, and then under neutral fairness preferences.

It can be seen that in IPD projects, when the risk aversion coefficient of participants increases, the risk they are willing to bear will decrease accordingly, so that the output utility and the profit-sharing coefficient also decrease accordingly, and the total revenue they obtained will also decrease. Secondly, in the initial state ($\rho = 3$), participants can obtain a high output utility when they have a vertical–horizontal fairness preference, so they can also obtain high total revenue under a vertical–horizontal fairness preference. With the reduction of the risk aversion coefficient, participants first attain a high sharing coefficient under the vertical fairness preference and subsequently under the neutral fairness preference. Therefore, with the reduction of the risk aversion coefficient, participants can obtain high total revenue under the vertical fairness preference and then under the neutral fairness preference.

## 4. Conclusions

This study aimed to improve the profit-sharing mechanism of IPD projects, promote the reasonable sharing of project profits among alliance members, enhance team collaboration, and facilitate the successful implementation of IPD projects. Based on Stackelberg game theory and considering the fairness preferences of the participants, a profit-sharing model for the IPD projects that utilize BIM technology under multidimensional fairness preferences was constructed and the optimal sharing coefficient was found. Meanwhile, the influence of the fairness preference intensity, effort cost coefficient, effort utility value, and risk aversion coefficient of the participants on their output utility, profit-sharing coefficient, and total revenue were analyzed under the multidimensional fairness preferences, and the results are as follows:

(1) Among the multidimensional fairness preferences of participants, when participants have vertical–horizontal fairness preferences, their output utility and total revenue are the highest, and the profit-sharing coefficient is only lower than that with vertical fairness preferences. When the participants have vertical fairness preferences, the

profit-sharing coefficient they obtain is the highest, while the output utility and total revenue are the lowest. When participants have horizontal fairness preferences, their output utility and total revenue are higher than those when they have neutral fairness preferences, while the profit-sharing coefficient is lower. Therefore, in IPD projects, participants can consider vertical–horizontal fairness preferences to improve the profits of the whole team and increase the profit-sharing coefficient obtained by themselves. At the same time, the owner should attach great importance to the fairness preference behavior of the participants when sharing the profits.

(2) In terms of fairness preference, when participants have vertical fairness preferences, their output utility, profit-sharing coefficient, and total revenue will increase with the increase of the preference intensity. When participants have horizontal fairness preferences, their output utility, profit-sharing coefficient, and total revenue will decrease with the increase of the preference intensity. When participants have vertical–horizontal fairness preferences, with the increase of the fairness preference intensity, the profit-sharing coefficient they obtain will decrease, and the output utility and total revenue will increase. Therefore, in the IPD projects, participants should adjust their own fairness preference intensity while ensuring the completion of contractual obligations so as to obtain satisfactory output utility, profit-sharing coefficients, and total revenue. At the same time, the owner needs to alleviate the fierce competition caused by the participants' horizontal fairness preference and mobilize the low effort enthusiasm caused by the participants with vertical fairness preferences so as to encourage them to achieve the target task.

(3) In terms of risk factors, the reduction of the risk aversion coefficient of participants will make them produce high output utility and obtain high sharing coefficients and total revenue. Therefore, the participants should reduce the risk level as much as possible and take the initiative to bear the project risk. At the same time, the owner should offer higher shared profits to the participants who undertake the high risk of the project as a return.

(4) In terms of effort factors, the increase in participants' effort utility value or the decrease in the effort cost coefficient both contribute to the enhancement of output utility, profit-sharing coefficients, and total revenue. Moreover, the improvement resulting from the increase in effort utility value is more substantial. Therefore, participants should strive to improve the utility value of unit effort and also reduce the cost. At the same time, the owner should give higher shared profits to the participants who improved the utility value of unit effort and realized the increase of project benefit.

In summary, in the profit-sharing mechanism of IPD projects using BIM technology, the owner should give great importance to the fair preference behaviors of the participants. The intense competition caused by horizontal fair preferences should be alleviated, and efforts to address the low initiative caused by vertical fair preferences should be stimulated. Additionally, the owner should provide substantial shared profits to participants who enhance the utility value of their unit efforts and contribute to increasing the overall project benefits, as well as those who bear high project risks, in order to motivate them to achieve the project objectives. The intensity of participants' fairness preferences should correspond to their own situations and the team's realities. Concurrently, participants should strive to enhance their risk resilience, proactively undertake project risks, and dedicate themselves to elevating unit effort utility value while reducing costs. While satisfying contractual obligations, participants should aim to enhance the interests of the entire project team, increase their own profits, and promote the realization of IPD project functionality or value.

In this paper, Stackelberg game theory was applied, taking into account the multidimensional fairness preferences of the participants. A profit-sharing model for IPD projects using BIM technology was constructed, which can provide support for the design of a fair and reasonable profit-sharing mechanism for IPD projects. Through simulation, it becomes possible to clarify the mechanisms underlying changes in behavioral choices between owners and participants, contribute to the development of more reasonable contracts and

collaboration arrangements by the owners, and ultimately maximize the return on investment for the projects. It benefits stakeholders in actively responding to project construction, optimizing their productivity, and efficiently managing and mitigating risks so as to ensure the unity and cooperation of all members and promote the smooth implementation of IPD projects. Simultaneously, it supplements the methods and applications in the realm of profit-sharing mechanisms.

The theoretical research in this paper has some limitations. Firstly, due to the complexity and persistence of the IPD projects, certain parameters such as project contract duration, cash discount rate, and other parameters were not considered in the construction of the model. Further refinements are necessary for subsequent research. Secondly, among the participants of IPD projects, the profit-sharing mechanism was limited to the owner, architect, and contractor. Future research could encompass a broader range of participants, including supervisors and banks. Additionally, this paper only considers IPD projects using multi-party contracts and does not address how to establish profit-sharing models if project participants use other IPD contracts. Finally, in the multi-party contract of the IPD project, other participants (consultants, suppliers, etc.) would separately sign subcontracting contracts with the aforementioned three parties. This paper did not consider the impact of these other participants on the decision-making of the aforementioned three key parties, and further discussion is needed in future research.

**Author Contributions:** Conceptualization and methodology, L.W.; writing—original draft preparation, M.T.; writing—review and editing, X.A.; software and visualization, G.D. All authors have read and agreed to the published version of the manuscript.

**Funding:** This research received no external funding.

**Data Availability Statement:** The data presented in this study are available on request from the corresponding author.

**Conflicts of Interest:** The authors declare no conflict of interest.

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
