# Peer review of "Research on Profit-Sharing Mechanism of IPD Projects Considering Multidimensional Fairness Preferences and BIM"

_systems, doi:10.3390/systems11090477_

Round 1
Reviewer 1 Report
This research proposes a profit-sharing mechanism appropriate to IPD projects implementing BIM practice. The topic is timely and addresses an important topic considering the fact that reasonable profit sharing is key to derive successful project outcomes by enhancing collaborations among a variety of stakeholders involved in IPD+BIM projects.
However, based on the current version of the manuscript, the research methodology is quite vague - first of all, it's unclear why specific modeling method (Stackelberg game theory) has been selected rather than other available ones. Why the selected method is a better choice in terms of enhancing the effectiveness, reliability, or accuracy of their findings? To address this issue, I’d recommend the authors strengthen the literature review section (introduction in the original version) by elaborating the different modeling methods with their pros and cons.
It would greatly enhance the understanding of the research methodologies if an overall research process could be visualized through a figure that presents the entire methodology.
Lines 120-130: the authors should describe how those assumptions are made, and also “Stackelberg game” must be elaborated to help readers understand the methods adopted in this research.
I am also curious how the outcomes of this research can benefit practitioners or non-academic readers for potential applications this research describes.
Table 1: please describe where the values listed in the table came from and how reliable those data are.
Overall, this paper can be significantly improved to make it reader-friendly. Especially, there needs to be a good grammatical review as there are many issues throughout the manuscript.
Author Response
Many thanks for the insightful comments and suggestions. We have made corresponding revision according to the suggestions. Words in blue are the changes we have made in the text. The following is the answers and revisions we have made in response to the questions and suggestions on an item by item basis.
RESPONSE TO REVIEWER # 1
Reviewer 1:
- Based on the current version of the manuscript, the research methodology is quite vague - first of all, it's unclear why specific modeling method (Stackelberg game theory) has been selected rather than other available ones. Why the selected method is a better choice in terms of enhancing the effectiveness, reliability, or accuracy of their findings? To address this issue, I’d recommend the authors strengthen the literature review section (introduction in the original version) by elaborating the different modeling methods with their pros and cons.
Response 1:
Thanks again for your comments and suggestions. Your advice will be the motivation for us to move on.
We've added detailed explanations of the different modeling approaches and their advantages and disadvantages. Please see lines 105-135. Words in blue are the changes we have made in the text. According to your valuable advice, we have made a lot of changes to this paper to make it more perfect. Thanks again.
- It would greatly enhance the understanding of the research methodologies if an overall research process could be visualized through a figure that presents the entire methodology.
Response 2:
We have incorporated a technology roadmap to visualize the research methodology process and provided a concise introduction to the research methodology. please refer to lines 234-245.
- Lines 120-130: the authors should describe how those assumptions are made, and also “Stackelberg game” must be elaborated to help readers understand the methods adopted in this research.
Response 3:
We have completed supplementation of the hypothesis. please refer to lines 247-266. Words in blue are the changes we have made in the text.
At the same time, we have provided a detailed description of the "Stackelberg game" to assist readers in understanding the methodology. please refer to lines 227-234.
- I am also curious how the outcomes of this research can benefit practitioners or non-academic readers for potential applications this research describes.
Response 4:
We have supplemented the practical significance of this study and explained how it can benefit practitioners and other personnel. please refer to lines 774-785.
This paper constructs the IPD project profit sharing model based on the participants' multidimensional fairness preference, which can provide support for the fair and reasonable IPD project revenue sharing mechanism design. Through simulation, the outcomes of this research contribute to clarifying the mechanism of changing behavioral choices between owners and participants, assisting owners in devising more rational collaborative arrangements to maximize project investment returns. It benefits all participants in actively responding to project construction, optimizing their own output utility, and better managing and mitigating risks. Thus, it ensures the unity and cooperation of all members and promotes the smooth implementation of IPD projects.
- Table 1: please describe where the values listed in the table came from and how reliable those data are.
Response 5:
We have supplemented the source of the simulated data. please refer to lines 499-502.
we incorporated IPD case studies gathered from AIA and established parameter settings by drawing upon the risk aversion coefficient and risk standard deviation from Guo et al. (2022), as well as the effort cost coefficient and effort utility value from Onur's pertinent research.
Reviewer 2 Report
Overall this is an interesting paper that develops a profit sharing model for BIM+IPD projects using game theory and considers multidimensional fairness preferences of participants. The model is logically constructed and the simulation provides some useful insights. However, there are a few areas that could be improved:
Introduction:
The introduction provides good background and motivation on the importance of profit sharing for BIM+IPD project success. The authors could expand a bit more on the key research gap this study aims to address.
The overall objective and highlights of the modeling approach could be stated more clearly upfront.
Literature Review:
The literature review covers relevant works on profit sharing factors, methods, and applications in IPD projects. To further strengthen it, the authors could specifically discuss any prior research that has looked at integrating fairness preferences into profit sharing models for construction projects. This would help situate the novelty of their approach.
Model Development:
The model is logically constructed using Stackelberg game theory considering multidimensional fairness preferences. The assumptions are reasonable.
When introducing the notation for the first time such as effort level ai, effort cost coefficient ki, etc., it would be helpful to clearly define each term.
In Section 2.2.4 when deriving the risk cost function, it is not fully clear how the mean, variance and risk cost expressions are obtained. More explanation or references would help.
Simulation Study:
The simulation results provide helpful insights into how different parameters affect the output utility, profit sharing and total revenue under different fairness preferences.
The authors could consider doing some validation of the model outputs against real project data if available. This would further demonstrate the practical applicability of the model.
In the conclusions, the authors could provide more specific guidance on how their model and findings can be applied by project stakeholders to improve profit sharing in BIM+IPD projects.
Overall this is a well-done modeling study on an important topic. Addressing the above suggestions would further strengthen the paper. The results provide valuable insights to enhance profit sharing mechanisms and promote successful BIM+IPD project delivery.
Minor editing of English language required
Author Response
Research on Profit Sharing Mechanism of IPD Projects Considering Multidimensional Fairness Preference and BIM
Many thanks for the insightful comments and suggestions. We have made corresponding revision according to the suggestions. Words in blue are the changes we have made in the text. The following is the answers and revisions we have made in response to the questions and suggestions on an item by item basis.
RESPONSE TO REVIEWER # 2
Reviewer 2:
Introduction:
- The introduction provides good background and motivation on the importance of profit sharing for BIM+IPD project success. The authors could expand a bit more on the key research gap this study aims to address.
Response 1:
Thanks again for your comments and suggestions. Your advice will be the motivation for us to move on.
We have further expanded the introduction, providing detailed explanation of the key research gaps that this study aims to address. Please see lines 184-195. Words in blue are the changes we have made in the text. According to your valuable advice, we have made a lot of changes to this paper to make it more perfect. Thanks again.
- The overall objective and highlights of the modeling approach could be stated more clearly upfront.
Response 2:
We have supplemented the overall objectives and highlights of the modeling approach. Please see lines 201-215.
The supplementary contents are also follows:
(1) Based on the Stackelberg game theory, with the project owner as the "leader" and the participants as the "followers" within the context of IPD mode, the mutual impact of decisions between the owner and participants is analyzed.
(2) Considering the application of BIM technology by participants and the impact of fairness preferences on the overall project revenue, the horizontal fairness preferences, vertical fairness preferences, and vertical-horizontal fairness preferences of participants are introduced to construct a profit sharing model for IPD projects.
(3) Through simulation, explore the impact of participants' fairness preference in-tensity, effort utility value, effort cost coefficient, and risk avoidance coefficient on their output utility, shared coefficient, and total revenue. The objective is to encourage project owner to establish rational cooperation arrangements and stimulate active engagement from all participants in project construction, ensuring cohesive collaboration among members and the smooth completion of IPD projects.
Literature Review:
- The literature review covers relevant works on profit sharing factors, methods, and applications in IPD projects. To further strengthen it, the authors could specifically discuss any prior research that has looked at integrating fairness preferences into profit sharing models for construction projects. This would help situate the novelty of their approach.
Response 3:
We have added several references that incorporate fairness preference into construction project profit sharing model. Please see lines 160-179.
Model Development:
- When introducing the notation for the first time such as effort level ai, effort cost coefficient ki, etc., it would be helpful to clearly define each term.
Response 4:
We have inserted a table before the model analysis and construction section to introduce all terms used in the paper, aiding in a clear understanding of each term. Please see line 280.
- In Section 2.2.4 when deriving the risk cost function, it is not fully clear how the mean, variance and risk cost expressions are obtained. More explanation or references would help.
Response 5:
We have provided further detailed explanation of the derivation of the risk cost function in Section 2.2.4. Please see lines 350-364.
Simulation Study:
- The authors could consider doing some validation of the model outputs against real project data if available. This would further demonstrate the practical applicability of the model.
Response 6:
The simulation in the manuscript was conducted using actual case, and the simulation results are consistent with actual scenarios. As such, this can to a certain extent validate the efficacy of the model. Moving forward, we will also actively strive to promote the practical application of the model in our future research and endeavors, refining the research findings based on the actual application outcomes.
- In the conclusions, the authors could provide more specific guidance on how their model and findings can be applied by project stakeholders to improve profit sharing in BIM+IPD projects.
Response 7:
We have supplemented specific guidance on profit sharing in IPD projects in the conclusion. Please see lines 760-773.
In the profit sharing mechanism of IPD projects, the owner should give great importance to the fair preference behaviors of the participants. The intense competition caused by horizontal fair preferences should be alleviated, and efforts to address the low initiative caused by vertical fair preferences should be stimulated. Additionally, the owner should provide substantial shared profits to participants who enhance their effort utility value, as well as those who bear high project risks, in order to motivate them to achieve the project objectives.
The intensity of participants' fairness preferences should correspond to their own situations and the team's realities. Concurrently, participants should strive to enhance their risk resilience, proactively undertake project risks, and dedicate themselves to elevating unit effort utility value while reducing costs.
Reviewer 3 Report
Dear Authors,
Thank you for presenting this study that offers valuable insights into integrating Building Information Modeling (BIM) and Integrated Project Delivery (IPD) to deal with profit sharing and proposes a game analysis model for profit sharing based on the Stackelberg game theory.
Please, find below my recommendations about the work.
General Comments
1. Profit sharing is an issue of IPD, not BIM. It is recommended to revise the manuscript title and approach to something like “Research on Profit Sharing Mechanism of IPD Projects Considering Multidimensional Fairness Preference and BIM.”
1. Introduction
2. Please, revise and add to the introduction “The Bible of IPD”: Fischer, M.; Ashcraft, H.; Reed, D.; Khanzode, A. Integrating Project Delivery; John Wiley & Sons: Hoboken, NJ, USA, 2017.
3. It is also recommended to give a broader view of BIM, which is VDC (Virtual Design and Construction). VDC is the engine to implement IPD. You can find further information about VDC here: Del Savio, A. A., Vidal Quincot, J. F., Bazán Montalto, A. D., Rischmoller Delgado, L. A., & Fischer, M. (2022). Virtual Design and Construction (VDC) Framework: A Current Review, Update, and Discussion. Applied Sciences, 12(23), 12178. MDPI AG. http://dx.doi.org/10.3390/app122312178
4. Please, add a last paragraph to the introduction to present how the subjects are organized and introduced in the next manuscript chapters: “2. Development of …”, “3. Simulation Analysis”, etc. The reader should know in advance what he will find in the paper and how it is organized.
2. Development of BIM+IPD Projects Profit Sharing Model
5. The profit sharing is dynamic throughout all the project components. Some components will be shared equally between all the parties, while others will not be shared or have different percentage distributions. Please address this issue in this section (model).
6. On the other hand, many of the options presented in the model are not open, i.e., they are defined and agreed upon under a contract. Please, address this issue when all the profit-sharing mechanisms and percentages are previously defined in a contract between the parties.
3. Simulation Analysis
7. Please, support the relevant parameters used for the simulation analysis.
8. What is the applicability and relation of the adopted parameters to a real project situation/case since the model is very sensitive to these parameters?
4. Conclusions
9. Please, address the contractual requirements. An “optimal sharing coefficient” may not comply with contractual obligations from the parties.
10. Based on the information presented in the paper, and the need for case studies, it is not possible to confirm that the model can support the design of the profit-sharing mechanism of IPD projects. It is recommended to present case studies of projects, applying this model to analyze the results.
11. Please, expand the future research recommendations to deal with contractual constraints, type of projects, and involving other participants like suppliers.
I hope to have helped with the recommendations above, and I look forward to your answer.
Best regards,
The Reviewer
Author Response
Research on Profit Sharing Mechanism of IPD Projects Considering Multidimensional Fairness Preference and BIM
Many thanks for the insightful comments and suggestions. We have made corresponding revision according to the suggestions. Words in blue are the changes we have made in the text. The following is the answers and revisions we have made in response to the questions and suggestions on an item by item basis.
RESPONSE TO REVIEWER # 3
Reviewer 3:
General Comments
- Profit sharing is an issue of IPD, not BIM. It is recommended to revise the manuscript title and approach to something like “Research on Profit Sharing Mechanism of IPD Projects Considering Multidimensional Fairness Preference and BIM.”
Response 1:
Thanks again for your comments and suggestions. Your advice will be the motivation for us to move on.
We have revised the manuscript title and approach to "Research on Profit Sharing Mechanism of IPD Projects Considering Multidimensional Fairness Preference and BIM." Please see the full manuscript. According to your valuable advice, we have made a lot of changes to this paper to make it more perfect. Thanks again.
Introduction
- Please, revise and add to the introduction “The Bible of IPD”: Fischer, M.; Ashcraft, H.; Reed, D.; Khanzode, A. Integrating Project Delivery; John Wiley & Sons: Hoboken, NJ, USA, 2017.
Response 2:
I appreciate your recommendation of this book. We have made further revisions and additions to the Integrated Project Delivery (IPD) mode in the introduction section. Please see lines 31-36.
- It is also recommended to give a broader view of BIM, which is VDC (Virtual Design and Construction). VDC is the engine to implement IPD. You can find further information about VDC here: Del Savio, A. A., Vidal Quincot, J. F., Bazán Montalto, A. D., Rischmoller Delgado, L. A., & Fischer, M. (2022). Virtual Design and Construction (VDC) Framework: A Current Review, Update, and Discussion. Applied Sciences, 12(23), 12178. MDPI AG. http://dx.doi.org/10.3390/app122312178.
Response 3:
I deeply appreciate your recommendation. We have completed the extended introduction to the BIM technology. Please see lines 40-54.
- Please, add a last paragraph to the introduction to present how the subjects are organized and introduced in the next manuscript chapters: “2. Development of …”, “3. Simulation Analysis”, etc. The reader should know in advance what he will find in the paper and how it is organized.
Response 4:
We have added a paragraph at the end of the introduction to provide an overview of the overall structure of the paper. Please see lines 216-225.
Development of BIM+IPD Projects Profit Sharing Model
- The profit sharing is dynamic throughout all the project components. Some components will be shared equally between all the parties, while others will not be shared or have different percentage distributions. Please address this issue in this section (model).
Response 5:
We have provided a brief introduction to the varying profit sharing approaches for different project components and indicated that the profit in this study is a portion that can be measured by efforts such as technical contributions, resource inputs, and risk undertakings. Please see lines 287-296.
- On the other hand, many of the options presented in the model are not open, i.e., they are defined and agreed upon under a contract. Please, address this issue when all the profit-sharing mechanisms and percentages are previously defined in a contract between the parties.
Response 6:
We have previously addressed and resolved the matter of many options in the model being defined and agreed upon based on contractual terms. Please see lines 271-279.
While many options presented in the model are defined and agreed upon based on contractual terms, such as the roles and responsibilities of the parties involved, risk and reward sharing, the principle of "symmetry between profits and risks," "balance between investment and return," and "equitable sharing" are still followed when determining the sharing ratios for benefit allocation. In cases where parameters for the parties involved undergo changes in the future, additional negotiations with the owner can still be conducted in accordance with the aforementioned principles.
Simulation Analysis
- Please, support the relevant parameters used for the simulation analysis.
Response 7:
We have supplemented the source of the simulated data. please refer to lines 499-502.
we incorporated IPD case studies gathered from AIA and established parameter settings by drawing upon the risk aversion coefficient and risk standard deviation from Guo et al. (2022), as well as the effort cost coefficient and effort utility value from Onur's pertinent research.
- What is the applicability and relation of the adopted parameters to a real project situation/case since the model is very sensitive to these parameters?
Response 8:
We have explained the applicability and relevance of the parameters to real project situations/cases. please refer to lines 472-494.
Conclusions
- Please, address the contractual requirements. An “optimal sharing coefficient” may not comply with contractual obligations from the parties.
Response 9:
In the process of model construction, we use the maximization of the owner's total profit as the objective function and the total profits of the participants as constraint conditions. Therefore, the "optimal sharing coefficients" derived from the model solution satisfy the intentions of both parties.
In the conclusion, to ensure that the "optimal sharing coefficients" align with the contractual obligations of the parties, we have added supplementary explanations and recommend that owners incentivize participants who may exhibit lower motivation due to vertical fairness preference. Participants can adjust the intensity of fairness preference, ensuring they meet contractual obligations. please refer to lines 760-773.
- Based on the information presented in the paper, and the need for case studies, it is not possible to confirm that the model can support the design of the profit-sharing mechanism of IPD projects. It is recommended to present case studies of projects, applying this model to analyze the results.
Response 10:
The simulation in the manuscript was conducted using actual case, and the simulation results are consistent with actual scenarios. As such, this can to a certain extent validate the efficacy of the model. Moving forward, we will also actively strive to promote the practical application of the model in our future research and endeavors, refining the research findings based on the actual application outcomes.
- Please, expand the future research recommendations to deal with contractual constraints, type of projects, and involving other participants like suppliers.
Response 11:
We have further expanded the future research recommendations to address the involvement of other participants such as contract constraints, project types, and suppliers. please refer to lines792-798.
Round 2
Reviewer 3 Report
Dear Authors,
Thank you for sending a revised version of the manuscript addressing my comments.
Just one remark: the Reference number [7] is from 2022, not 2020.
Best regards,
The Reviewer